# Bipolar Membrane Electrodialysis for Direct Conversion of L-Ornithine Monohydrochloride to L-Ornithine

**DOI:** 10.3390/ijms241713174

**Published:** 2023-08-24

**Authors:** Jinfeng He, Wenlong Liu, Jianrong Hao, Xixi Ma, Zhiyi Zheng, Yinghan Fang, Yuxin Liang, Zhihao Tian, Li Sun, Chuanrun Li, Haiyang Yan

**Affiliations:** 1Pharmaceutical Engineering Technology Research Center, School of Pharmacy, Anhui University of Chinese Medicine, Hefei 230012, China; jfhe@ahtcm.edu.cn (J.H.); lwlyeah@163.com (W.L.);; 2Anhui Province Key Laboratory of Pharmaceutical Preparation Technology and Application, Hefei 230012, China

**Keywords:** bipolar membrane electrodialysis, L-ornithine monohydrochloride, L-ornithine, conversion, separation mechanism

## Abstract

In this study, bipolar membrane electrodialysis was proposed to directly convert L-ornithine monohydrochloride to L-ornithine. The stack configuration was optimized in the BP-A (BP, bipolar membrane; A, anion exchange membrane) configuration with the Cl^−^ ion migration through the anion exchange membrane rather than the BP-A-C (C, cation exchange membrane) and the BP-C configurations with the L-ornithine^+^ ion migration through the cation exchange membrane. Both the conversion ratio and current efficiency follow BP-A > BP-A-C > BP-C, and the energy consumption follows BP-A < BP-A-C < BP-C. Additionally, the voltage drop across the membrane stack (two repeating units) and the feed concentration were optimized as 7.5 V and 0.50 mol/L, respectively, due to the low value of the sum of H^+^ ions leakage (from the acid compartment to the base compartment) and OH^−^ ions migration (from the base compartment to the acid compartment) through the anion exchange membrane. As a result, high conversion ratio (96.1%), high current efficiency (95.5%) and low energy consumption (0.31 kWh/kg L-ornithine) can be achieved. Therefore, bipolar membrane electrodialysis is an efficient, low energy consumption and environmentally friendly method to directly convert L-ornithine monohydrochloride to L-ornithine.

## 1. Introduction

L-ornithine is an important medicinal intermediate and is acknowledged to be an effective antidote to ammonium in blood [1,2]. For instance, L-ornithine L-aspartate is an effective ammonia-lowering agent in hepatic encephalopathy [3,4,5]. Therefore, L-ornithine has been used widely in pharmaceutical industries in recent years, and the demand is growing steadily owing to the increasing number of people affected by liver diseases worldwide. In practice, L-ornithine can be produced by extraction method [6], chemical synthesis method [7], microbial fermentation method [1], and enzymatic method [8], etc. As a basic amino acid, L-ornithine is generally produced in the form of L-ornithine monohydrochloride (L-ornithine·HCl). However, L-ornithine·HCl needs to be converted to L-ornithine to prepare the relevant medicines such as L-ornithine L-aspartate [2]. As for the dissociation equilibrium, L-ornithine has three dissociation constants (pK_1_ = 1.71, pK_2_ = 8.69, pK_3_ = 10.76) [9], and its various ionic forms are shown in Figure 1.

Therefore, L-ornithine would be in the form of L-ornithine^2+^, L-ornithine^+^, L-ornithine^±^ and L-ornithine^−^ at different surrounding pHs. According to the ionization equilibrium, the mole fraction of these four specifications can be calculated by Equations (1)–(4):(1)$$\delta_{L\text{-}ornithine^{2+}}=\frac{1}{1+10^{pH-pK_{1}}+10^{2pH-pK_{1}-pK_{2}}+10^{3pH-pK_{1}-pK_{2}-pK_{3}}}$$
(2)$$\delta_{L\text{-}ornithine^{+}}=10^{pH-pK_{1}}\cdot \delta_{L\text{-}ornithine^{2+}}$$
(3)$$\delta_{L\text{-}ornithine^{\pm }}=10^{2pH-pK_{1}-pK_{2}}\cdot \delta_{L\text{-}ornithine^{2+}}$$
(4)$$\delta_{L\text{-}ornithine^{-}}=10^{3pH-pK_{1}-pK_{2}-pK_{3}}\cdot \delta_{L\text{-}ornithine^{-}}$$
where $\delta_{L\text{-}ornithine^{2+}}$, $\delta_{L\text{-}ornithine^{+}}$, $\delta_{L\text{-}ornithine^{\pm }}$, and $\delta_{L\text{-}ornithine^{-}}$ are the fraction ratios of L-ornithine^2+^, L-ornithine^+^, L-ornithine^±^, and L-ornithine^−^, respectively. As a result, the mole fractions of various forms of L-ornithine at different surrounding pHs are shown in Figure 1.

In the traditional conversion process, L-ornithine·HCl solution is firstly passed through the cylinder of strongly acidic ion exchange resin (H-type) [2], and the positively charged L-ornithine (L-ornithine^+^) can be absorbed in the resin. After that, the L-ornithine^+^ is effused by aqueous ammonia from the resin [2]. Lastly, the free L-ornithine (L-ornithine^±^) can be obtained from the collected effluent after the ammonia removal process [2]. These processes not only consume large amounts of fresh chemicals, but also produce a large amount of wastewater discharged by elution and resin regeneration processes. Therefore, it is urgent to develop an efficient and environmentally friendly method to produce L-ornithine.

Electrodialysis (ED) is an electro-driving membrane process, in which cation exchange membrane (CEM or C) and anion exchange membrane (AEM or A) are alternatively arranged between the anode and cathode [10]. Cations in the dilute solution migrate through CEM with a direction from the anode to the cathode under the driving force of direct current, while anions in the dilute solution migrate through AEM with the opposite direction [10]. Currently, ED has been used widely in the fields of salt concentration [11], feed desalination [12,13], ion metathesis reaction [14,15], ion substitution [16,17], acid and base production [18,19,20,21,22], etc. For instance, as for the separation of basic amino acid, Kattan Readi et al. [23] reported that 1,5 pentanediamine (PDA) can be separated from L-arginine (Arg) by ED at a pH of 10 (PDA^+1.5^/Arg^±^) with a high recovery ratio of 63%. Wang et al. [24] reported the separation of L-lysine from mixed amino acids (L-glutamic and L-lysine) by bipolar membrane electrodialysis (BMED) using self-prepared porous CEM, in which the L-lysine can be separated efficiently with a high rejection ratio of L-glutamic (~100%) and the energy consumption is 6.43–9.53 kWh/kg. When it comes to the conversion of basic amino acids, Aghajanyan et al. [20] reported electromembrane transformation of L-lysine monohydrochlorides into their zwitterionic form by two- and four-chamber ED, in which current efficiencies were 75.9% and 73.1%, respectively. Zhang et al. [16] reported the recovery of L-lysine from L-lysine monohydrochloride by ion substitution electrodialysis (ISED, C-A-A configuration), in which the removal ratio of Cl^−^ reached to >95%, but the current efficiency was as low as 20.5%, and the energy consumption was relatively high (9.0 kWh/kg) at the current density of 5–15 mA/cm^2^. Similarly, Kumar et al. [17] reported the conversion of lysine monohydrochloride to lysine by ISED (A-A configuration) with the electrolysis reaction, in which the recovery ratio of lysine was 96.2%, the energy consumption was 2.07 kWh/kg, and the current efficiency was 93.2% at the current density of 10 mA/cm^2^. Furthermore, Zhang et al. [21] reported the production of L-lysine from L-lysine monohydrochloride by BMED (the BP-A-C configuration, BP-bipolar membrane), in which the removal ratio of Cl^−^ was 86.6%, the current efficiency was 24%, and the energy consumption was as high as 28.2 kWh/kg at the certain conditions. Herein, water dissociation occurs inside the BPM, and the generated OH^−^ ions migrate through the cation exchange layer of the BPM to the feed compartment [25]. Simultaneously, Cl^−^ ions in the feed compartment migrate through AEM to the acid compartment, thus the L-lysine was produced. Eliseeva et al. [22] reported the recovery and concentration of basic amino acids (lysine, arginine and histidine) by BMED (the BP-C configuration), in which the positively charged basic amino acids in the acid compartment migrate through CEM to the base compartment, and combine with OH^−^ ions generated by BPM to produce the basic amino acids. Meanwhile, the concentration of basic amino acids can be increased 35–50 times. Therefore, ED-based membrane processes, especially for BMED, are efficient and feasible for the production of basic amino acids.

Hence, in this work, the conversion of L-ornithine·HCl to L-ornithine was carried out by BMED process. Based on the above reports, we can find that different stack configurations of BMED have different separation mechanisms for the conversion of L-ornithine·HCl to L-ornithine. The ions migration can be classified in two ways: (1) Cl^−^ ions migration through AEM from the base compartment to the acid compartment and (2) positively charged L-ornithine migration through CEM from the acid compartment (or salt compartment) to the base compartment. Also, co-ion migrations in these two ways are different, which will influence the separation performances. Furthermore, the co-ion migration can be directly influenced by operating parameters such as the applied voltage drop (or current density) and feed concentration [26]. Therefore, in this study, three kinds of configurations (BP-A, BP-C, and BP-A-C configurations) are applied to produce L-ornithine. Separation mechanisms for these three configurations were clarified, and separation performances were evaluated. After the optimization of membrane stack configuration, operating parameters such as the applied voltage drop (or current density) and feed concentration were investigated to optimize the separation performances in consideration of conversion ratio, current efficiency, and energy consumption.

## 2. Results and Discussion

### 2.1. Effect of the Stack Configuration

As shown in Figure 2 and Figure 3, different stack configurations have different ion migration mechanisms, resulting in the different separation performances shown in Figure 4. For the case of the BP-C configuration, the feed solution (500 mL 0.5 mol/L L-ornithine·HCl solution) was fed into the acid compartment and the L-ornithine^+^ ions in the acid compartment migrated through CEM to the base compartment under the driving force of the direct current. In the base compartment, L-ornithine^+^ reacted with the OH^−^ ions dissociated by BPM to produce L-ornithine (or L-ornithine^±^), which is in accordance with the value of pH of the base compartment as shown in Figure 4a. This is because L-ornithine is mainly in the form of L-ornithine^±^ at the pH of ~10 (Figure 1). Figure 4b shows that the $C_{L\text{-}ornithine}$ increases slowly at the final stage of batch experiment; the reason can be ascribed to the migration of H^+^ ions from the acid compartment to the base compartment. Generally, L-ornithine^+^ ions are transported through CEM by Stokesian mechanism [27], while H^+^ ions are transported by Grotthuss mechanism [28]. Furthermore, from Table 1, we can see that the bare ion radius of L-ornithine^+^ (3.17 Å) is much larger than that of H^+^ (0.28 Å). Therefore, H^+^ ions are more easily transported through CEM compared with L-ornithine^+^ ions. Meanwhile, the pH of the acid solution decreases gradually as a function of time (Figure 4a), in other words, the concentration of HCl in the acid compartment increases gradually. Therefore, in the final stage of the batch experiment, the concentration of L-ornithine^+^ in the acid compartment decreases a lot, and more H^+^ ions competed with L-ornithine^+^ ions to migrate to the base compartment.

For the case of the BP-A-C configuration, the feed solution was fed into the salt compartment, then the L-ornithine^+^ ions migrate through CEM to the base compartment, and reacted with OH^−^ ions, which were dissociated by the BPM, to produce L-ornithine; the Cl^−^ ions migrated through AEM to the acid compartment and combined with H^+^ ions generated by BPM to produce HCl Figure 4b shows that $C_{L\text{-}ornithine}$ increases gradually to 0.42 mol/L, higher than that of the BP-C configuration (0.38 mol/L). Figure 4a shows that the pH of acid solution in the case of the BP-A-C configuration is lower than that in the case of the BP-C configuration, which means that HCl can be recovered more efficiently in the BP-A-C configuration (Appendix A). In addition, it is interesting that the pH of the salt solution decreases after 80 min, which is due to the co-ion (H^+^ ion) transport through AEM from the acid compartment to the salt compartment, lowering the pH of the salt solution.

For the case of the BP-A configuration, the feed solution was fed into the base compartment, then the L-ornithine^+^ ions could be reacted with OH^−^ ions, which were generated by BPM, to produce L-ornithine, and the Cl^−^ ions were migrated through AEM to the acid compartment. Figure 4a shows the pH of the base solution increases gradually from ~6 to ~10, which means that the L-ornithine·HCl in the base solution is continuously converted to L-ornithine. Figure 4b shows that the $C_{L\text{-}ornithine}$ can be increased to as high as 0.50 mol/L, which is higher than that in the above two configurations. Figure 4a and Appendix A also show that HCl can be efficiently recovered in the acid compartment due to the decreasing pH of the acid solution or the increasing concentration of HCl. In addition, Appendix A shows that the conductivity of the base solution decreases from 29.70 mS/cm to 2.75 mS/cm as a function of time, the overall value of which is obviously higher than that in the case of BP-C (0–1.55 mS/cm) and BP-A-C (0–2.07 mS/cm) configurations. In addition, the conductivity of the acid solution increases dramatically from 0 to 158.2 mS/cm. Both high conductivities of acid and base solutions mean the low resistance of the membrane stack, resulting in a high current density as shown in Figure 4c, which further shortens the running time of batch experiment (25 min).

When it comes to the conversion ratio, it can be calculated according to $C_{L\text{-}ornithine}$. Figure 4d shows that the conversion ratio of three configurations follows the order of BP-A (96.5%) > BP-A-C (91.2%) > BP-C (79.7%). As for the current efficiency, it can be calculated from the results shown in Figure 4b. The higher $C_{L\text{-}ornithine}$ is, the higher current efficiency when the batch experiment consumed the same amount of charge. Therefore, the current efficiency of three configurations follows the order of BP-A (83.6%) > BP-A-C (77.7%) > BP-C (44.0%). At last, the energy consumptions were calculated, which follow the order of BP-A (0.41 kWh/kg) < BP-A-C (0.90 kWh/kg) < BP-C (1.48 kWh/kg).

Overall, the stack configuration was optimized as a BP-A configuration with Cl^−^ ion migration rather than L-ornithine^+^ ion migration in the BP-C and BP-A-C configurations. For the optimized configuration, the running time of batch experiment is only 25 min; the $C_{L\text{-}ornithine}$, the conversion ratio and the current efficiency can reach to as high as 0.5 mol/L, 96.5% and 83.6%, respectively; and the energy consumption is as low as 0.41 kWh/kg L-ornithine.

### 2.2. Effect of the Voltage Drop across the Membrane Stack

As mentioned in the above section, the stack configuration was optimized as a BP-A configuration, and the current efficiency was 83.6%. The reason for the relatively low current efficiency can be ascribed to two aspects: (1) the migration of OH^−^ ions from the base compartment to the acid compartment, and (2) the co-ion leakage of H^+^ ions [32,33] from the acid compartment to the base compartment. On the one hand, compared with Cl^−^ ion, OH^−^ ion has a smaller hydrated radius as shown in Table 1, and can easily migrate through AEM to the acid compartment. On the other hand, H^+^ ion has a smaller hydrated radius and a higher diffusion coefficient, resulting in the co-ion leakage [32] through AEM under the high concentration of HCl in the acid compartment. The above two aspects can be influenced directly by the driving force, i.e., voltage drop or current density. Therefore, various voltage drops across the membrane stack (5 V, 7.5 V, and 10 V) were investigated. The feed solution was 500 mL 0.50 mol/L L-ornithine·HCl solution. Each batch operation was stopped once the conductivity of acid compartment has no obvious increase, as can be seen in Appendix A.

Figure 5a shows that the running time of the batch experiment can be shortened as the voltage drop increased, because a high voltage drop can result in high current density (Figure 5c). For instance, the current density has a low value of 0–5.6 mA/cm^2^ at the voltage drop of 5 V, which increases to 0–31.8 mA/cm^2^ as the voltage drop increases to 7.5 V, and increases to the maximum value (52.9 mA/cm^2^, 10 A) of the DC power as the voltage drop increases to 10 V. Herein, in the case of 10 V, after the current density reached to the maximum value, the constant voltage drop mode was converted to the constant current density mode, and the voltage drop exhibited a decreasing trend due to the increasing conductivity of the acid solution (Appendix A). After that, the conductivity of the base solution decreased to a much lower value (Appendix A), resulting in an increase in voltage drop. Subsequently, the voltage drop was returned to 10 V again, and the current density decreased. Based on the time and current density, the changing trend of $C_{L\text{-}ornithine}$ for various voltage drops as a function of charge (time multiplied current) was shown in Figure 5b. It is interesting that, in the case of 7.5 V and 10 V, the $C_{L\text{-}ornithine}$ could reach the high value of 0.50 mol/L, which is higher than that of 5 V (0.46 mol/L). Herein, high $C_{L\text{-}ornithine}$ means a high conversion ratio. Figure 5d shows that high conversion ratios of 96.1% and 96.5% were achieved for the voltage drops of 7.5 V and 10 V, respectively, while the conversion ratio was as low as 88.5% for the low voltage drop of 5 V.

Another interesting phenomenon is that, in the case of 7.5 V, $C_{L\text{-}ornithine}$ increased the most rapidly compared with the other two cases (high voltage drop of 10 V and low voltage drop of 5 V) as a function of charge. In addition, the $C_{L\text{-}ornithine}$ in the case of 5 V increased more slowly than that in the case of 10 V at the final stage of experiment. Similarly, the changing trend of $C_{HCl}$ in the acid compartment is shown in Appendix A. A previous report [32] has indicated that a higher H^+^ ion leakage through AEM from the acid compartment to the adjacent compartment was obtained at a lower current density and vice versa. Therefore, in the case of 5 V, the reason why more charges consumed compared with the case of 7.5 V should be ascribed to the H^+^ ions leakage through AEM from the acid compartment to the base compartment. As the voltage drop increases to 10 V, on the contrary, the H^+^ ions leakage through AEM would be lower than that in the case of 7.5 V. However, Figure 5b shows that more charge was consumed in the case of 10 V compared with that in the case of 7.5 V. Herein, the cause should be ascribed to the large amounts of OH^−^ ions (generated by BPM) which migrated through AEM from the base compartment to the acid compartment instead of Cl^−^ ions migration in the case of 10 V. Therefore, the sum of H^+^ ions leakage (from the acid compartment to the base compartment) and OH^−^ ions migration (from the base compartment to the acid compartment) through AEM in the case of 7.5 V is lower than that in the case of 10 V, and that in the case of 5 V is the highest.

Based on the above results, the current efficiencies were calculated and shown in Figure 5d. We can see that the current efficiencies for the three cases follow the order of 7.5 V (95.5%) > 10 V (83.6%) > 5 V (69.9%). This is because the lower the value of the sum of H^+^ ions leakage and OH^−^ ions migration through AEM, the higher the current efficiency is. As for the energy consumption, in general, the higher the voltage drop or current density, the higher the energy consumption is [18]. Therefore, the energy consumption in the case of 10 V is as high as 0.41 kWh/kg. However, the energy consumption in the case of 5 V (0.32 kWh/kg) is slightly higher than that of the case of 7.5 V (0.31 kWh/kg), the main reason should be attributed to the highest value of the sum of H^+^ ions leakage and OH^−^ ions migration through AEM.

Overall, the voltage drop was optimized as 7.5 V in consideration of the high $C_{L\text{-}ornithine}$ (0.50 mL/L), high conversion ratio (96.1%), high current efficiency (95.5%), and low energy consumption (0.31 kWh/kg).

### 2.3. Effect of the Feed Concentration

As mentioned above, the current efficiency can be influenced by voltage drop or current density due to the migration of OH^−^ ions from the base compartment to the acid compartment and the co-ion leakage of H^+^ ions from the acid compartment to the base compartment. In fact, the OH^−^ ions migration and H^+^ ions leakage can also be influenced by the feed concentration because the current density is influenced by the feed concentration under a certain voltage drop. In addition, high feed concentration means high H^+^ ions leakage in the conversion process, especially in the latter stage of the batch experiment. Therefore, the feed concentration (0.25, 0.50, 0.75, and 1.00 mol/L) was investigated with the optimized voltage drop of 7.5 V. Each batch operation was stopped once the conductivity of the acid compartment had no obvious increase, as can be seen in Appendix A.

Figure 6a shows that more running time was needed for the batch experiment at higher feed concentrations. Figure 6c shows that high feed concentrations can result in high current density due to the high conductivities of acid and base solution as shown in Appendix A. Then, the charge of batch experiment can be calculated according to the running time and current density, and the relation between $C_{L\text{-}ornithine}$ and charge for various feed concentrations ($C_{feed}$) is shown in Appendix A. We can see that high $C_{L\text{-}ornithine}$ can be achieved for high $C_{feed}$, but there are no obvious correlations referring to the OH^−^ ions migration and H^+^ ions leakage for various $C_{feed}$. Further, the ratios of $C_{L\text{-}ornithine}$ to $C_{feed}$ ($C_{L\text{-}ornithine}$/$C_{feed}$) for various feed concentrations as a function of the ratio of charge to $C_{feed}$ (charge/$C_{feed}$) were calculated as shown in Figure 6b. Clearly, $C_{L\text{-}ornithine}$/$C_{feed}$ increases with increasing charge/$C_{feed}$, but the $C_{L\text{-}ornithine}$/$C_{feed}$ in the case of 0.50 mol/L is of the highest value at the same charge/$C_{feed}$. This means that the sum of OH^−^ ions migration and H^+^ ions leakage in the case of 0.50 mol/L is the lowest, which is advantageous for the conversion process. The $C_{L\text{-}ornithine}$/$C_{feed}$ in the case of 0.25 mol/L is slightly lower than that of the case of 0.50 mol/L, the reason for this is the high H^+^ ions leakage, since the current density of the case of 0. 25 mol/L is lower than that of the case of 0.50 mol/L. As the $C_{feed}$ increases from 0.50 mol/L to 1.00 mol/L, the $C_{L\text{-}ornithine}$/$C_{feed}$ exhibits a decreasing trend, the reason should be ascribed to both the increased OH^−^ ions migration and the increased H^+^ ions leakage. Because, on the one hand, the current density increases as the $C_{feed}$ increases, resulting in an increase in OH^−^ ions migration as aforementioned; on the other hand, the concentration of HCl increases as the $C_{feed}$ increases (Appendix A), resulting in an increase in H^+^ ions leakage [32].

Figure 6d shows the conversion ratio, current efficiency, and energy consumption. We can see that all the conversion ratios for various $C_{feed}$ can reach to >95%. The current efficiency follows the order of 0.5 mol/L (95.5%) > 0.25 mol/L (89.0%) > 0.75 mol/L (78.9%) > 1.00 mol/L (73.5%), which is in accordance with that of $C_{L\text{-}ornithine}$/$C_{feed}$. The energy consumption of the case of 0.25 mol/L is relatively high, which is due to the high resistance of the acid and base solutions as shown in Appendix A. Because of the higher the solution resistance, the higher the energy lost [34]. As the $C_{feed}$ increases from the 0.25 mol/L to 0.50 mol/L, the resistance of the acid and base solutions decreases and the $C_{L\text{-}ornithine}$/$C_{feed}$ increases, resulting in a decrease in the energy consumption (from 0.38 kWh/kg to 0.31 kWh/kg). As the $C_{feed}$ increases from 0.50 to 1.00 mol/L, the $C_{L\text{-}ornithine}$/$C_{feed}$ decreases, resulting in an increase in the energy consumption (from 0.31 kWh/kg to 0.34–0.35 mol/L). Overall, the feed concentration was optimized at 0.50 mol/L in consideration of the high current efficiency (95.5%) and low energy consumption (0.31 kWh/kg).

## 3. Materials and Methods

### 3.1. Materials

The membranes used in the experiment are listed in Table 2, in which the bipolar membrane is BP-1E (Tokuyama Co., Tokyo, Japan), and the CEM and AEM are CIS and AIS (Shandong Tianwei Membrane Technology Co., Ltd., Weifang, China), respectively. L-ornithine monohydrochloride (L-ornithine·HCl) was purchased from Shanghai Macklin Biochemical Co., Ltd. (Shanghai, China), and sodium sulfate (Na_2_SO_4_) was purchased from Sinopharm Chemical Regent Co., Ltd. (Shanghai, China). Deionized water was used.

### 3.2. Experimental Setup

As shown in Figure 2 and Figure 3, the experimental setup has a DC power supply (HSPY-100-10, Beijing Hanshengpuyuan Science and Technology Co., Ltd., Beijing, China), a membrane stack (CJED-1020, Hefei Chemjoy Polymer Materials Co., Ltd., Hefei, China), some peristaltic pumps (BT600L-YT15, Baoding Lead Fluid Technology Co., Ltd., Baoding, China) for recirculating the corresponding solutions, and some tanks such as an acid tank, a base tank, and a salt tank, as well as an electrolyte tank. For the DC power supply, the deviations in the output voltage drop and direct current were ±0.1% + 20 mV and ±0.5% + 20 mA, respectively. Specifically, the membrane stacks with two repeating units were assembled in BMED setup. In the membrane stack, the membranes were alternately arranged between the anode and cathode made of titanium coated with ruthenium, forming two compartments in each repeating unit for BP-A and BP-C compartments (Figure 2), and forming three compartments in each repeating unit for the BP-A-C configuration (Figure 3). The adjacent membranes were separated by a spacer with a thickness of 0.75 mm. The effective area of each membrane was 189 cm^2^. Each solution was pumped from the tank into the corresponding compartment via a pump and the flow rate was controlled at 320 mL/min (the liner flow velocity 4 cm/s). All experiments in the BMED process were carried out in the batch mode with a constant stack voltage drop.

### 3.3. Experimental Procedure

In this study, the effects of the stack configuration, the applied voltage drop across the membrane stack, and the feed concentration of L-ornithine·HCl solution were systematically investigated. Firstly, three kinds of configurations, i.e., the BP-A, BP-C, and BP-A-C configurations, were applied to convert the L-ornithine·HCl to the L-ornithine. For the BP-A and BP-C configurations, each membrane stack was equipped with three tanks including an electrolyte tank, a base tank, and an acid tank. In the case of the BP-A configuration, the acid tank was filled with 500 mL DI water, and the base tank was filled with 500 mL 0.5 mol/L L-ornithine·HCl solution (pH: 5–6). In contrast, in the case of the BP-C configuration, 500 mL DI water and 500 mL 0.5 mol/L L-ornithine·HCl solution were filled into the base tank and the acid tank, respectively. For the BP-A-C configuration, the membrane stack was equipped with four tanks including an electrolyte tank, a base tank (500 mL DI water), a salt tank (500 mL 0.5 mol/L L-ornithine·HCl solution), and an acid tank (500 mL DI water). In each experiment, 500 mL 0.3 mol/L Na_2_SO_4_ was filled into the electrode tank, and the applied voltage drop across the membrane stack was 10 V. Secondly, various voltage drops across the membrane stack (5, 7.5, and 10 V) were applied to the membrane stack using the optimized configuration, and the BMED performances were evaluated. Lastly, the effect of feed concentration (0.25, 0.50, 0.75 and 1.00 mol/L L-ornithine·HCl solution) on BMED performances was investigated under the optimized configuration and voltage drop. All the experiments were conducted at room temperature.

### 3.4. Sample Analysis and Data Calculation

#### 3.4.1. Sample Analysis

During the experiment, the pH and conductivity of various solutions were monitored by a pH meter with a deviation of ±0.002 pH (ST5000, OHAUS Instruments, Parsippany, NJ, USA) and a conductivity meter with a deviation of ±1.0%FS (DDBJ-350F, Shanghai INESA and Scientific Instrument Co., Ltd., Shanghai, China), respectively. In the case of the BP-A-C and BP-A configurations, HCl could be recovered in the acid tank, and the concentration of the recovered HCl was titrated by NaOH standard solution with the phenolphthalein as an indicator, where the experimental data were collected through three independent measurements. On the contrary, in the case of the BP-C configuration, the generated HCl and L-ornithine·HCl were mixed in the acid compartment, thus HCl could not be efficiently recovered, and the concentration of HCl was not measured. For the BP-A-C and BP-C configurations, L-ornithine could be recovered in the base compartment, and the concentration of the recovered L-ornithine in the base tank was calculated according to the chemical oxygen demand (COD) values measured by multi-parameter water quality tester with a deviation within ±5% (5B-3B, Beijing Lianhua YongXing Science and Technology Development Co., Ltd., Beijing, China). But in the case of the BP-A configuration, L-ornithine·HCl and the produced L-ornithine were present in the same compartment (base compartment). According to the mass conservation, the molar quantity of the produced L-ornithine was equal to that of the Cl^−^ ions migrated from the base compartment to the acid compartment. Therefore, $C_{L\text{-}ornithine,t}$ in the base compartment can be calculated as follows:(5)$$C_{L\text{-}ornithine,t}=\frac{C_{HCl,t}V_{acid,t}}{V_{base,t}}$$
where $C_{L\text{-}ornithine,t}$ (mol/L) is the concentration of the recovered L-ornithine at time *t*, $C_{HCl,t}$ is the concentration of HCl in acid compartment at time *t*, $V_{acid,t}$ is the volume of acid solution at time *t*, and $V_{base,t}$ is the volume of base solution at time *t*.

#### 3.4.2. Conversion Ratio, Current Efficiency and Energy Consumption

The conversion ratio (*CR*, %) of L-ornithine was calculated as:(6)$$CR=\frac{C_{L\text{-}ornithine,t}V_{base,t}}{C_{L\text{-}ornithine\cdot HCl,0}V_{base,0}}\times 100\%$$
where $C_{L\text{-}ornithine\cdot HCl,0}$ (mol/L) is the initial feed concentration of L-ornithine·HCl, and $V_{base,0}$ (L) is the initial volume of base solution.

The current efficiency (*CE*, %) can be calculated as [36]:(7)$$CE=\frac{C_{L\text{-}ornithine,t}V_{base,t}F}{\int_{0}^{t}NIdt}\times 100$$
where *N* is the number of repeating units (*N* = 2), *I* is the applied current (A), and F is the Faraday constant (96,485 C/equiv.).

The energy consumption (*EC*, kWh/kg) of L-ornithine converted from L-ornithine·HCl can be calculated as [36]:(8)$$EC=\frac{\int_{0}^{t}UIdt}{C_{L\text{-}ornithine,t}V_{base,t}M}$$
where *U* is the applied voltage drop across membrane stack (*V*) and M is the molar weight of L-ornithine (132.2 g/mol).

## 4. Conclusions

In this study, L-ornithine monohydrochloride was directly converted to L-ornithine by the BMED process. Firstly, the stack configurations including BP-C, BP-A-C, and BP-A were investigated to clarify the separation mechanism of the BMED process. The results indicate that the BP-A configuration with Cl^−^ ion migration through AEM is more preferable than the other two configurations with L-ornithine^+^ ion migration through CEM, due to the high conversion ratio, high current efficiency, and low energy consumption. Specifically, both the conversion ratio and current efficiency follow BP-A > BP-A-C > BP-C, and the energy consumption follows BP-A < BP-A-C < BP-C. Secondly, the effect of voltage drop (5, 7.5 and 10 V) across the membrane stack with the BP-A configuration on BMED performances was investigated. The results indicate that the voltage drop was optimized as 7.5 V due to the lowest value of the sum of H^+^ ions leakage (from the acid compartment to the base compartment) and OH^−^ ions migration (from the base compartment to the acid compartment) through AEM. Meanwhile, the high conversion ratio, high current efficiency, and low energy consumption can be achieved. Lastly, the effect of feed concentration (0.25–1.00 mol/L) was investigated, and the relation between $C_{L\text{-}ornithine}$/$C_{feed}$ and charge/$C_{feed}$ was calculated to further elaborate the H^+^ ions leakage and OH^−^ ions migration. The results indicate that the feed concentration was optimized at 0.50 mol/L due to the lowest value of the sum of H^+^ ions leakage and OH^−^ ions migration. Overall, at the optimized conditions, a high conversion ratio (96.1%), a high current efficiency (95.5%) and a low energy consumption (0.31 kWh/kg) can be achieved. Therefore, BMED is an efficient, low energy consumption, and environmentally friendly method to directly convert L-ornithine monohydrochloride to L-ornithine. Further studies are required to model the ion transfer through ion exchange membranes [26,37] to improve the separation performances, and for the scale-up application and membrane fouling in treating the real feed solution containing impurities.

## Data Availability

The data presented in this study are available on request from the corresponding author.

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
