# Peer review of "Bipolar Membrane Electrodialysis for Direct Conversion of L-Ornithine Monohydrochloride to L-Ornithine"

_ijms, 2023, doi:10.3390/ijms241713174_

Round 1
Reviewer 1 Report
The article is useful for the practice of bipolar membrane electrodialysis and suitable for publication in the special issue. The authors use commercial polymer ion-exchange membranes (bipolar and monopolar) and study EDBM recovery of a basic amino acid (L-ornithine) from its hydrochloride solution applying different configurations of the stack and choosing the most effective one. The influence of feed concentration and driving force (voltage drop) on the L-ornithine conversion ratio and current efficiency as well as on the energy consumption is discussed for the chosen configuration (BP-A).
To my opinion, it is strongly advisable to include additional references into Introduction section because there are interesting articles in literature describing the application of electrodialysis with bipolar membranes for the basic amino acids separation and concentration. The authors have mentioned only one work of Y. Zhang et al. (2012) in this field.
The standard deviations for all measurements (COD, pH, conductivity) are not indicated by the authors. The concentration of chlorides is not measured directly in this study or the data are not included.
There are some incorrect terms used by the authors. For example, it is not possible to name basic amino acid as “alkaline” amino acid (line 35, 72). Besides, I recommend to make some changes in Table 1. It is not correct to use term “Molecular formula” for Cl-, H+, OH-.. Molecular weight Mr (for ornithine) should be nondimensional. Molar weight (M) has the dimension g/mol.
As per Scheme 1 in the manuscript it is incorrect to consider and designate ornithine like molecule Orn0. Ornithine exists in the form of bipolar ion and should be shown as bipolar ion having + and – charges as well. The bipolar structure is confirmed by high dipole moment.
Misprints: lines 85 (from ?), 281.
There are some drawbacks regarding language. Please, check lines 48-50, 65-66, 70-71, 72, 112-113, 125, 135-136, 142, 146, 166-167, 180, 207, 218, 265-267, 268, 278-280, 318, 332. English should be improved.
Reviewer 2 Report
Transport of polyvalent ions in electrodialysis, while important for wastewater valorisation and food processing, is a challenge both for description of underlying phenomena and for interpretation of experimental results. In my opinion the authors successfully dealt with the problem and obtained the sound results. My only major concern lays in novelty, since the experience of working with other polyvalent ions says that if the process allows avoiding the transfer of polycharged ions through the membrane, then it will be the most efficient route, since this route will not deal with protonation and deprotonation, chelate-like interaction, pH dependence of the composition and, essentially, the nature of main charge carriers, with differences in ionic radius and so forth. The main conclusion was that the configuration in which amino acid is not transported through the membrane is the best one, as might be expected. However, the presented results are important as new data on the transport of polyvalent species even while it is not about some major breakthrough.
There are several minor things that authors might be interested in ironing out.
1) at several points (see for example lines 120 and 181) it is implied that the protons are transported faster than the amino acid cations due to differences in ionic radius. While it obviously is so, the authors might also want to point out that there even the mechanisms of ion transport are different, and while amino acid ions are transported by stokesian mechanism, protons are transported by Grotthuss mechanism.
2) if I were the authors I would give the chemical structure of amino acid earlier in the manuscript, and would also note the type of membrane matrix there, since it might be important to understand from the beginning if the amino acid is acidic, alkaline or neutral, and also to know if it is aromatic or not. For example, phenylalanine has stacking interactions with aromatic matrices.
3) it seems that the abstract has too many abbreviations, less formal rephrase might improve readability.
4) in some instances I am unsure about the English wording. See for example first row in table 1, is "molecular formula" term used for ions? I also recall the recommendation to use "potential difference" or "potential drop" in place of "voltage". Can the authors please check again?
Reviewer 3 Report
Dear Authors, please see the attached file

English could be improved. I have no strict comments
Reviewer 4 Report
In the present manuscript “Bipolar membrane electrodialysis for direct conversion of L-ornithine monohydrochloride to L-ornithine”, the authors propose a process for extracting L-ornithine from L-ornithine.HCl by bipolar membrane electrodialysis (BMED) process. They were inspired from another study on the L-lysine monohydrochloride with the same process. Two essential parts are identified in this manuscript: first, the choice of the best BMED configuration system, and then the optimization of membrane stack configuration.
The subject is in line with the aims of the International Journal of molecular Science. The authors have limited themselves to the essentials and haven't gone into too much detail. The manuscript is well written and presented. It can be accepted with minor revisions, and after answering these two points:
How many times were these experiments repeated? Are the given values averages or the best values obtained? What are the standard deviations and errors of these values?
Did the authors observe any drift over time that might be due to membrane fouling? If so, what did they do to overcome this problem?
Round 2
Reviewer 1 Report
The improvement of Introduction is insufficient. The authors have not added the references (and their discussion) which are close to the topic of the article - electromembrane conversion of basic amino acid hydrochloride into basic amino acid. The authors have written about few reports concerning conversion of L-ornithine hydrochloride to L-ornithine by BMED process (lines 92-93). However, there are no the following references in the text of the manuscript and their consideration. There are also close articles concerning other basic amino acids hydrochlorides conversion (Aghajanyan A.E., Tsaturyan A.O. et al, Shaposhnik V.A. et al., Zhang X., Lu W. etc.). You can consider some of them to show the importance and novelty of this study.
At the same time there are references in the manuscript which are far from the text they should confirm.
The examples.
Lines 62-64. It is obvious that references 11 and 12 are not an origin of the principle of conventional ED. The reference 13 is more suitable as a review article.
Line 67. Reference 14 is misleading here. Common information should have a reference to a book or a review article.
Line 89. References 24-25 are very interesting and new but these articles refer to the improvement of BPMs using special catalyzers. The authors do not use them, they write about the principle of EDBM performance. Reference 26 is a review article. To my opinion, it is OK here.
The similar situation with references 28 and 29. The Grotthuss mechanism of proton transport was not proposed in 2019-2021.
References 36 and 37 are not suitable references for membranes properties.
Reference 39 in the Section “Conclusion” is devoted to neutralization dialysis process..
Table 1. Instead of “Ionic formula” I recommend to include “Properties of the considered ions”. Also add, please, “Hydrated ion radius”.
Line 42. Scheme 1. I recommend its title? for example, "Various ionic forms of ornithine". We can not write reactions because the authors do not show protons in the scheme.
Line 45. The applied equations are well known and common so it is not necessary to make reference to the own article 10.
The improvement of language is insufficient.
Line 35 (add “…is produced in the form…”), lines 64-65, 121-122,147 replace “can be migrated” by “migrate”, lines 87-89, 99-102… – bad English Grammer, lines 127, 135 – replace “latter stage” by “final stage”, line 132 replace “transport” by “transported” ,
Lines 148,158, 236 – use “…OH-ions generated by BPM…”
Please, check all the text.
